# Viscosity-Sensitive Solvatochromic Fluorescent Probes for Lipid Droplets Staining

**DOI:** 10.3390/bios12100851

**Published:** 2022-10-09

**Authors:** Mao-Hua Wang, Wei-Long Cui, Yun-Hao Yang, Jian-Yong Wang

**Affiliations:** State Key Laboratory of Biobased Material and Green Papermaking, Key Laboratory of Paper Science and Technology of Ministry of Education, Faculty of Light Industry, Qi Lu University of Technology, Shandong Academy of Sciences, Jinan 250353, China

**Keywords:** lipid droplets, viscosity-sensitive, polarity-sensitive, solvatochromism

## Abstract

Lipid droplets (LDs) are simple intracellular storage sites for neutral lipids and exhibit important impact on many physiological processes. For example, the changes in the polar microenvironment inside LDs could affect physiological processes, such as lipid metabolism and storage, protein degradation, signal transduction, and enzyme catalysis. Herein, a new fluorescent chemo-sensor (Couoxo-LD) was formulated by our molecular design strategy. The probe could be applied to effectively label intracellular lipid droplets. Intriguingly, Couoxo-LD demonstrated positive sensitivity to both polarity and viscosity, which might be attributed to its D-π-A structure and the twisted rotational behavior of the carbon–carbon double bond (TICT). Additionally, Couoxo-LD was successfully implemented in cellular imaging due to its excellent selectivity, pH stability, and low biotoxicity. In HeLa cells, the co-localization curve between Couoxo-LD and commercial lipid droplet dyes overlapped at 0.93. The results indicated that the probe could selectively sense LDs in HeLa cells. Meanwhile, Couoxo-LD can be applied for in vivo imaging of zebrafish.

## 1. Introduction

Lipid droplets (LDs) serve as simple intracellular storage sites for neutral lipids and consist of a non-polar neutral lipid core [1]. Studies have shown that LDs are not only simple energy stores but also complex and dynamic multifunctional organelles [2]. For example, alterations in the polar microenvironment surrounding LDs affect physiological processes, such as lipid metabolism and storage, protein degradation, signal transduction, and enzyme catalysis [3]. Furthermore, previous studies reported that homeostasis of the LDs microenvironment was associated with diseases, such as obesity, cardiovascular disease, and diabetes mellitus [4,5,6]. Cancer cells exhibit a strong affinity for fatty acids and cholesterol, which are over-stored in lipid droplets. [7]. The high levels of LDs in tumors provide a potential means of monitoring and treating cancer [8]. Therefore, the tracking and monitoring of LDs are essential. In addition, LDs consist a unique structure (a single phospholipid membrane with a hydrophilic “head (group)” facing the cytoplasm and an internal storage of lipid cores, such as triglycerides and cholesterol esters [9,10]), i.e., a hydrophobic and viscous environment inside the lipid droplets. Theoretically, if the probe exhibits different fluorescence properties in these two microenvironments, they could be used as a tool for labeling LDs [11,12].

Fluorescence imaging has established itself as a beneficial tool for studying biological systems because of its high sensitivity, accessibility, non-invasiveness, and real-time and in situ detection of target molecules [13,14,15,16,17,18,19]. So far, several fluorescent probes for specifically imaging LDs have been reported [20,21,22,23,24,25,26,27]. For example, Yu et al., designed two heteroindole-based two-photon fluorescent probes and visualized the polarity of LDs at a cellular level and in zebrafish larvae [15]. However, there were only a few probes that were reported to respond positively to both the polarity and viscosity of lipid droplets. Therefore, there is a requirement to develop a strong color-changing LDs probe for sensing both polarity and viscosity [28,29].

Since Perkin et al. first synthesized artificial coumarins by chemical synthesis in 1868 [30], coumarin derivatives have been widely used in the design of small-molecule fluorescent sensors because of their good biocompatibility, controlled structure, and stable and strong fluorescence emission intensity [31,32,33,34,35,36,37,38,39,40,41]. Here, a solvent–chromogenic lipid droplets fluorescent probe, Couoxo-LD, was synthesized using benzoylglycine and coumarin fluorescent moiety. Couoxo-LD consisted an oxazolone and a coumarin derivative linked by a double bond. In optical characterization tests, the emission wavelength of the probe exhibited a significant red shift with increasing solvent polarity. The emission intensity of the probe increased with increasing solvent viscosity, showing regular polarity-viscosity-sensitive characteristics. The properties of good biocompatibility and pH stability were expressed in this probe. In addition, Couoxo-LD exhibited satisfactory lipid droplets targeting, possessed a high degree of overlap with commercial lipid droplets dye co-localization imaging, and had been successfully applied to cells and zebrafish imaging.

## 2. Synthesis of Probes

### 2.1. Reagents and Materials

The materials and reagents involved in the experiments were obtained commercially; furthermore, no secondary purification was carried out. The instruments used in the experiments were described in detail in the supporting materials.

### 2.2. Synthesis of Couoxo-LD

The synthesis of compound **1** was reported in detail in earlier work [42]. The synthetic design route of Couoxo-LD was shown in Figure 1 and the specific synthesis was referred to in Ref. [43].

Compound **1** (7-(diethylaMino)-2-oxo-2H-chroMene-3-carbaldehyde) (49.1 mg, 0.2 mmol), compound **2** (benzoylglycine) (39.42 mg, 0.22 mmol), and triphenylphosphine (5.25 mg, 0.022 mmol) were dissolved in anhydrous acetic anhydride (54.14 mg 50 μL, 0.53 mmol) and stirred for 4 h at 130 °C. After completion of the reaction (thin-layer chromatography monitoring), reaction mixture was cooled to room temperature. Ethyl alcohol (95%, 5.0 mL) was added to produce a large amount of precipitation. The product was obtained by filtration and recrystallization from ethanol (66.81 mg, 86 %). ^1^H NMR (400 MHz, CDCl_3_) δ 9.17 (s, 1H), 8.13–8.07 (m, 2H), 7.63–7.57 (m, 1H), 7.56–7.49 (m, 3H), 7.43 (d, *J* = 8.9 Hz, 1H), 6.64 (dd, *J* = 9.0, 2.4 Hz, 1H), 6.41 (d, *J* = 2.4 Hz, 1H), 3.45 (q, *J* = 7.1 Hz, 4H), 1.25 (t, *J* = 7.1 Hz, 6H). ^13^C NMR (101 MHz, CDCl_3_) δ 166.74, 162.76, 161.37, 157.09, 152.31, 146.86, 133.16, 131.94, 131.40, 128.94, 128.14, 125.67, 125.27, 113.79, 110.03, 109.86, 97.34, 45.30, 12.52.

## 3. Result and Discussion

### 3.1. Probe Design and Discussion

As mentioned above, LDs are distinguished from the surrounding viscous environment due to their unique internal structure, namely a phospholipid monolayer and a lipid core composed of fatty acids. The large amount of water and inorganic ions as components of the cytoplasmic solute provides a polar environment around LDs. Meanwhile, the characteristic environments such as viscosity and non-polarity exhibited inside LDs serve as a reliable theory to support our development of efficient lipid droplets fluorescent probes.

Based on the above principles, we designed and constructed a fluorescent probe Couoxo-LD with a structure sensitive to both viscosity and polarity. Couoxo-LD, consisting of coumarin derivatives and benzoylglycine, possessed strong electronic push–pull properties. In addition, the coumarin scaffold was selected as the monomeric component of Couoxo-LD, giving the probe a certain lipophilicity and therefore a higher sensitivity to lipid droplets. By introducing a diethylamine electron donor, Couoxo-LD was designed as a molecule with a typical D-π-A structure. In conclusion, the solvent-altering effect of Couoxo-LD was the main reason for illuminating intracellular lipid droplets. In an aqueous environment, such as a cytoplasmic solute, the twisted rotational behavior (TICT) around the carbon–carbon double bond hindered the emission behavior of Couoxo-LD [44]. In contrast to this, in non-polar media, such as LDs, it would release a strong signal through the locally excited (LE) state. That is, the vinyl structure in Couoxo-LD allowed the free rotation of the probe molecule that was restricted by the environment, which affected the emission behavior of the probe (Figure 1). The probe Couoxo-LD was characterized by ^1^H NMR and ^13^C NMR (Appendix A).

### 3.2. Study of Photophysical Properties of Probes

The molecular probe that possessed a donor (D)-π-acceptor (A) structure exhibited a pronounced solvatochromic effect and its photophysical properties varied with solvent polarity. Therefore, the absorption and emission behaviors of Couoxo-LD in different polar solvents were investigated (Figure 1 and Appendix A), such as 1,4-dioxane, methanol (MeOH), dichloromethane (DCM), N,N-dimethylformamide (DMF), ethanol (EtOH), tetrahydrofuran (THF), toluene (Tol), dimethyl sulfoxide (DMSO), ethyl acetate (EtOAc), acetone, and acetonitrile (CH_3_CN).

As shown in Figure 1 and Appendix A, the spectral data of the probe in different polar solvents were firstly explored, and Couoxo-LD showed strong absorption and emission phenomena and exhibited red-shifted behavior. From the emission peak at 576 nm in toluene to 622 nm in dimethylsulfoxide, the emission peak of Couoxo-LD underwent a red shift of about 50 nm. Under 365 nm-UV irradiations, it was clearly seen that this probe with an electronic push–pull structure exhibited a clear solvent discoloration effect accompanied by a change in fluorescence color from blue (toluene) to orange-red (dimethylsulfoxide) (Figure 1a). The above presented results demonstrate the ICT effect of Couoxo-LD. As mentioned above, we evaluated the environment-sensitive and alike-solvent-discoloration effect of the Couoxo-LD by studying the emission behavior of Couoxo-LD under different solvent polarities. In particular, Couoxo-LD exhibited a strong positive solvent discoloration effect, which was consistent with the closely reported fluorophore [45]. Moreover, a gradual increase in the polarity parameter E_T_(30) from 33.9 kcal^−1^ to 55.4 kcal^−1^ molar concentration resulted in a continuous red shift of the maximum emission wavelength of the probe (Figure 1a). More significantly, we found that Couoxo-LD showed a satisfactory linear relationship between the emission peak in different polar solvents and the E_T_(30) of the solvent with their Pearson correlation coefficient of 0.991 (Figure 1b and Appendix A) and a slope of 4.1 nm E_T_(30) units. This was caused by the function of the intra-molecular charge transfer effect (ICT) from the electron-giving N,N-diethyl unit to the electron-accepting oxazol-5(4H) one. These results indicated that the Couoxo-LD photophysical properties were closely related to solvent polarity.

To verify the above conjecture, we performed density general function theory (DFT) calculations for Couoxo-LD, and based on them we further optimized the electron cloud around Couoxo-LD using the Gaussian’09 program and DFT-derived Multiwfn and VMD program models. The distribution of Couoxo-LD in the more polar PBS-buffered solvent and the distribution of HOMO and LUMO in the less polar dioxane solvent were also calculated (Figure 2). The results show that the energy band gap ΔE of Couoxo-LD in aqueous solution is smaller than that in dioxane solvent. Obviously, the larger the energy band gap ΔE, the smaller the wavelength of the maximum absorption peak because the energy required for the electron leap is large. It was further verified that the wavelength of the absorption peak of Couoxo-LD in a low-polarity solvent environment (dioxane) was smaller than that of the absorption peak in a high-polarity solution (PBS buffer).

Since the Φ_F_ of Couoxo-LD depended strongly on the solvent’s properties (Table 1), the Φ_F_ increased with increasing solvent polarity (Toluene to Acetone) and reached a maximum. The increase in Φ_F_ due to charge transfer (negative solvatokinetic effect) could be explained by several mechanisms, such as proximity effects and conformational changes. The decrease in Φ_F_ from DMSO to DMF (positive solvatokinetic effect) could be attributed to strong ICT interactions. As described in the literature [46], the fluorescence quantum yield of Couoxo-LD depended on the polarity of the solvent, as well as on specific solute–solvent interactions, such as hydrogen bonding and high photostability.

### 3.3. The Emission Behavior of the Probe in PBS and Dioxane

Furthermore, we investigated the spectroscopic testing of the probe in different volume ratios of PBS buffer–dioxane mixtures (Figure 3). From the above experiments, we discovered that the emission behavior of the probe was related to the polarity of the medium; therefore, we further explored the lipid solubility test of the probe in different polar environments. In the binary solvent system of PBS buffer and dioxane, increasing the proportion of PBS buffer (f_w_) from 40% to 100%, the fluorescence intensity of Couoxo-LD showed a trend of increasing and then rapidly decreasing. Its emission peak underwent a certain red shift, which we judged to be caused by the increased polarity of the solvent mixture. The emission of the probe decreased appreciably with the decrease of the lipid-soluble solvent. This might be due to the increasing content of the PBS buffer, in which the increased polarity of the solvent made Couoxo-LD more sensitive, masking the solubility of the probe to the lipid solvent. This results in a reduction in the emission intensity of Couoxo-LD and a red shift of the maximum emission peak. All these findings suggested that the optical properties of Couoxo-LD were closely related to the environmental polarity.

### 3.4. Probe Emission Behavior in Viscous Environments

We verified the viscosity-emitting behavior of Couoxo-LD in binary systems with different ratios of PBS buffer and glycerol. As shown in Figure 4a, the emission intensity gradually increased with the increase of glycerol content. We further explored the relationship between the probe in a medium with different volume ratios of PBS buffer–Glycerol and the maximum emission intensity. The emission intensity of Couoxo-LD increased exponentially with the viscosity of the system (Figure 4b), which was consistent with the D-π-A structure of the probe molecule we mentioned earlier. In a low-viscosity environment, the free intra-molecular rotation leads to a slackening of the excitation energy, which greatly attenuated the emission phenomenon. In high-viscosity environments, free rotation within the probe molecule was inhibited by the environment and the molecule released energy primarily by radiation, resulting in a significant enhancement of fluorescence emission from Couoxo-LD. Furthermore, the color change of Couoxo-LD in the mixed system was found to be affected by polarity in the study (Figure 4c,d). This could be due to the higher polarity of the solution in the pure-PBS-buffer environment. With the addition of glycerol, the polarity in the mixed system decreased. A shift from red to yellowish green occurred, while the spectrum underwent a blue shift. Finally, when the glycerol ration was increased to 100%, the polarity of the system was greater than the mixed polarity of both and a red shift in the spectrum occurred and the color changed back to red. These results showed that Couoxo-LD could successfully monitor changes in viscosity and respond to polarity.

### 3.5. Probe Stability Study

To verify whether the Couoxo-LD was adapted to the sophisticated environment inside living cells, we investigated the emission behavior of Couoxo-LD in phosphate buffer solutions over a wide range of pH values. The results showed that the fluorescence intensity of Couoxo-LD decreased slightly when decreasing the solution pH (Figure 5a). Overall, the fluorescence intensity of Couoxo-LD was observed to be almost identical in different pH environments, demonstrating that the probe was unaffected by pH. We then investigated the response of the probe Couoxo-LD under different interfering ion environments (Figure 5b). The ions and different small biomolecules were added to PBS buffer and dioxane solutions, respectively, and the results showed that the fluorescence intensity of the probe did not change significantly in the organic solvent dioxane and PBS buffer, indicating that the emission behavior of Couoxo-LD was almost unaffected by the other chemicals. Furthermore, the emission intensity of Couoxo-LD in PBS buffer remained almost unchanged after continuous irradiation with a 365 nm-UV lamp for 1000 s, while it remained at a high level in dioxane, although it decreased slightly (Figure 5c). The excellent photostability indicated that the probe Couoxo-LD possessed a strong resistance to photobleaching and photo bursts.

### 3.6. Probe Couoxo-LD for Bioimaging Applications

The cytotoxicity of the probe was tested by the standard MTT method (Appendix A). Cells survival remained above 90% under incubation with a concentration of 20 μM Couoxo-LD, indicating that the cytotoxicity of Couoxo-LD was low and exhibited no significant impact on the cells testing. Encouraged by the excellent optical testing of the Couoxo-LD, we then estimated its cells imaging capabilities by laser scanning confocal (LSC) imaging techniques (Figure 6). BODIPY 493/503, a commercially available probe, was used for monitoring LDs as a control. We evaluated the absorption and emission spectra of Couoxo-LD and BODIPY (Appendix A). Living HeLa cells were stimulated with oleic acid in order to generate more LDs. Figure 6d demonstrates the co-localization images of both dyes on intracellular lipid droplets. As anticipated, Couoxo-LD labeled intracellular lipid droplets well (Figure 6e,f). Furthermore, the Pearson’s correlation coefficient between Couoxo-LD and the commercial dye BODIPY was as high as 0.93. These results suggested that Couoxo-LD possessed a good ability to sense LDs in living HeLa cells.

Meanwhile, to investigate the response of the probe to intracellular polarity, we performed imaging tests on the probe. In the control group, HeLa cells were imaged after incubation with culture medium for 30 min. In the experimental group, the cells were further treated with a preconfigured H_2_O_2_ (500 μM, 20 μL) solution for 20 min on the basis of control cells. The H_2_O_2_ solution would kill the cells, leading to a decrease in the number of intracellular lipid droplets and resulting in a change in intracellular polarity. At this point, the fluorescence of the green channel diminished from bright green (Figure 7b,f), the fluorescence of the red channel was lighted up (Figure 7c,g), and the images of the combined channels also showed a change from green to orange fluorescence (Figure 7d,h).

Zebrafish have similar digestive systems to humans, such as the liver and intestines. Additionally, their methods of digestion and nutrient absorption transport are highly similar to humans. Therefore, using live zebrafish to model intestinal lesions could help to further study human-related diseases. Approximately 70% of the zebrafish yolk sac fraction is neutral lipid, and by utilizing the lipid-specific fluorescent staining of the zebrafish yolk sac (Figure 8b and Appendix A), it was clearly observed that both dyes stained the zebrafish yolk sac. This indicates that Couoxo-LD can successfully label lipid droplets. Therefore, Couoxo-LD was informative for further studies on human physiology and pathology caused by abnormal expression of lipid droplets.

## 4. Conclusions

In this work, we designed a polar viscosity-sensitive fluorescent probe for targeting LDs. The probe Couoxo-LD was extremely sensitive to the polarity and viscosity of different media and showed intense fluorescence in LDs. In addition, the probe possessed splendid selectivity, low biotoxicity, and photostability. By artificially altering the external environment, Couoxo-LD could be competently used for discriminating changes in LDs polarity between living and post-mortem HeLa cells. Couoxo-LD was successfully applied for zebrafish imaging. Furthermore, lipid droplets co-localization imaging illustrated the accurate targeting of intracellular lipid droplets by Couoxo-LD. We believe that Couoxo-LD could be a powerful tool to study the processes associated with LDs.

## Data Availability

Not applicable.

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
