# Peer review of "Viscosity-Sensitive Solvatochromic Fluorescent Probes for Lipid Droplets Staining"

_biosensors, 2022, doi:10.3390/bios12100851_

Round 1
Reviewer 1 Report
Wang and co-workers developed a novel fluorescent probe Couoxo-LD with excellent selectivity, pH stability, and low biotoxicity for lipid droplets (LDs) sensing. The authors have well characterized the developed Couoxo-LD probe by 1H and 13C NMR and evaluated its photophysical properties (absorption and fluorescence) in different solvents. Next, the solvent polarity and viscosity of Couoxo-LD in different solvents or PBS/glycerol mixtures were investigated and supported by DFT calculations. Furthermore, the authors applied their probe to monitor not only changes in LDs polarity in live Hela cells and postmortem Hela cells, but also in zebrafish by confocal imaging. The Couoxo LD probe developed here will enable far-reaching applications to better understand the important role of LDs in biological processes. Overall, the results presented here are solid and supported by the conclusions.
I would recommend the publication of this interesting work in the journal of Biosensors after considering the minor revisions.
In revision, the following comments and corrections should be considered.
1. In the abstract, the chemo sensor is a single word, so please change as chemosensor.
2. In the abstract, line no 20, “The indicating” should be “This indicating”
3. In line no 83, please provide the yield units as 66.81 mg (66.81, 86 %).
4. In the caption of Figure 3, please add the title for a) and b) to make this clear.
5. Which standard was used to calculate the fluorescence quantum yield for the Couoxo LD and includes the experimental method with cited reference.
6. Per the journal data requirements, the authors should provide the HRMS data for Couoxo-LD derivatives synthesized in this manuscript. If the compounds have already been reported, please cite the relevant references.
7. Please include the 13C NMR data values in the manuscript.
8. ALL the Figure labels were small, please change the labels with an enlargement.
Author Response
Oct. 3, 2022
Dear reviewer:
Thank you very much for giving us the opportunity to revise our manuscript entitled " A novel polar viscosity-sensitive and strong solvochromic fluorescent probe and its application in lighting up lipid droplets"(biosensors-1936421) as an article in the Biosensors, we are also grateful for the reviewers’ comments.
We have completely revised the manuscript in response to your and the reviewers' comments. The changes in the revised manuscript have been highlighted electronically with a green background.
The list of changes or comments is shown as following:
Recommendation: Minor Revision
Comments:
This is an interesting new molecule and it has been characterised thoroughly. There are only a few minor things to address.
- Throughout there are a few key typos. For example, some missing units in section 2 (g/mg?), a missing dash in the NMR, occasional capital letters in the middle of sentences etc.
Reply: Thanks for reviewer’s valuable comments. Based on your suggestions, we added the missing units and symbols in section 2, corrected the occasional capital letters in the middle of the sentence. Similar errors that had occurred and 13C NMR data were added. For example, the final yield of the product, (66.81 mg, 86%), 1H NMR (400 MHz, CDCl3) δ 9.17 (s, 1H), 8.13–8.07 (m, 2H), etc., Please see paragraph 2, section 2.2 in the revised manuscript. Thank you again for your positive and constructive suggestions on our manuscript!
- Please give amount of triphenyl phosphine added to the reaction also.
Reply: Thanks for reviewer’s valuable comments. Based on your suggestions, we have added the amount of triphenylphosphine in the experimental part of the manuscript and checked for similar errors and reversals in the manuscript. Specifically, triphenylphosphine (5.25 mg, 0.022 mmol). Please see line 74, paragraph 2.2, page 2 in the revised manuscript. Thank you again for your positive and constructive suggestions on our manuscript!
- Figure 1 seems to need a caption – I wasn’t sure why these figures/tables were in their own section?
Reply: Thanks for reviewer’s valuable comments. Based on your suggestions, we have removed Figure 1 from the manuscript. It is a graphic abstract that summarizes the research and main innovations of the article. We have uploaded it in a separate file. Thank you again for your positive and constructive suggestions on our manuscript!
- Is the glycerol experiment impacting viscosity only or also polarity? I felt the explanation of these competing effects and control studies could be clearer.
Reply: Thanks for reviewer’s valuable comments. Based on your suggestions, we reworked and added to the competing effects and control studies in the glycerol experiments. The corrections and additions in the manuscript are as follows,we first verified the viscosity-emitting behavior of Couoxo-LD in binary systems with different ratios of PBS buffer and glycerol. As shown in Figure 4a, the emission intensity gradually increased with the increase of glycerol content. We further explored the relationship between the probe in a medium with different volume ratios of PBS buffer-Glycerol and the maximum emission intensity. The emission intensity of Couoxo-LD increased exponentially with the viscosity of the system (Figure 4b). This is consistent with the structure of the probe molecule we mentioned earlier, a molecule with a D-π-A structure. In a low viscosity environment, the free intra-molecular rotation leads to a slackening of the excitation energy, which greatly attenuates the emission phenomenon. In high-viscosity environments, free rotation within the probe molecule is inhibited by the environment and the molecule releases energy primarily by radiation, resulting in a significant enhancement of fluorescence emission from Couoxo-LD [50]. Furthermore, the color change of Couoxo-LD in the mixed system was found to be affected by polarity in the study (Figure 4c,d). This could be due to the higher polarity of the solution in the pure PBS buffer environment. With the addition of glycerol, the polarity in the mixed system decreased. A shift from red to yellowish green occurred, while the spectrum underwent a blue shift (Figure 4a,c). Finally, when glycerol was added to 100%, the polarity of the system was greater than the mixed polarity of both and a red shift in the spectrum occurred and the color changed back to red (Figure 4a,c). These results show that Couoxo-LD can successfully monitor changes in viscosity and respond to polarity. Please see line 220, paragraph 3.2 page 8 in the article. Thank you again for your positive and constructive suggestions on our manuscript!
- Can the authors improve the contrast in Fig 8g - it’s very faint vs the black control.
Reply: Thanks for reviewer’s valuable comments. We acquired fluorescence emission from Couoxo-LD by laser confocal fluorescence microscopy (Leica SP8) on different channels. The process of cell death induces an increase in the polarity of the microenvironment. In the experimental group cells were chosen to be treated with H2O2 to cause changes in the internal cellular environment. The results showed that the alteration of the intracellular polar microenvironment was weak, which may be responsible for the poor contrast effect of the red channel in the test results. Thank you again for your positive and constructive suggestions on our manuscript!
Jian-Yong Wang, Ph.D
Associate Professor
School of Light Industry and Engineering,
Qi Lu University of Technology (Shandong Academy of Sciences),
Jinan, Shandong 250353, P. R., China
Email: wjy@qlu.edu.cn
Reviewer 2 Report
This is an interesting new molecule and it has been characterised thoroughly. There are only a few minor things to address.
Throughout there are a few key typos. For example some missing units in section 2 (g/mg?), a missing dash in the NMR, occasional capital letters in the middle of sentences etc.
Please give amount of triphenyl phosphine added to the reaction also.
Figure 1 seems to need a caption – I wasn’t sure why these figures/tables were in their own section?
Is the glycerol experiment impacting viscosity only or also polarity? I felt the explanation of these competing effects and control studies could be clearer.
Can the authors improve the contrast in Fig 8g – it’s very faint vs the black control.
Author Response
Oct. 3, 2022
Dear reviewer:
Thank you very much for giving us the opportunity to revise our manuscript entitled " A novel polar viscosity-sensitive and strong solvochromic fluorescent probe and its application in lighting up lipid droplets"(biosensors-1936421) as an article in the Biosensors, we are also grateful for the reviewers’ comments.
We have completely revised the manuscript in response to your and the reviewers' comments. The changes in the revised manuscript have been highlighted electronically with a green background.
The list of changes or comments is shown as following:
Recommendation: Major Revision
Comments:
The manuscript by Wang et al. describes the using of a new fluorescent dye which is viscosity and polarity sensitive and has been studied in the context of lipid droplet staining, both in vitro and in vivo. While this new dye appears to have great utility in such applications and could be an important advancement in the field. One of key issues in the development of a fluorescent dye for lipid droplets is that the dye should enable multimodal imaging together with existing two commercially available dyes—BODIPY 493/503 and Nile Red. I like to recommend its publication after some decisive and major revisions commented below.
- 1. Authors should remove the word “Novel” or change it to “New” and remove the word “Strong” from the title. It better to revise the title like “Viscosity-sensitive Solvatochromic Fluorescent Probes for Lipid Droplets Staining’
Reply: Thanks for reviewer’s valuable comments. Based on your suggestions, we have corrected the word “Novel” to “New”, and changed the title to “Viscosity-sensitive Solvatochromic Fluorescent Probes for Lipid Droplets Staining”. Thank you again for your positive and constructive suggestions on our manuscript!
- Solvatochromic compounds generally shows the high quantum yields in nonpolar solvent and low quantum yields in polar solvents. But in table 1 the trend is opposite ΦFin Toluene is 5.2% while in DMSO and DMF is ~11%. Authors should explain and describe in the manuscript?
Reply: Thanks for reviewer’s valuable comments. Based on your suggestions, we have added the explanation and clarification of ΦF in Table 1. In response to the phenomenon that ΦF of Couoxo-LD in DMSO and DMF is 11%, while ΦF in toluene is 5.2%. Since the ΦF of Couoxo-LD strongly depends on the solvent properties (Table 1). The ΦF increases with increasing solvent polarity (Toluene to Acetone) and reaches a maximum. The increase in ΦF (negative solvatokinetic effect) due to charge transfer can be explained by several mechanisms, such as proximity effects and conformational changes. The decrease in ΦF from DMSO to DMF (positive solvatokinetic effect) can be attributed to strong ICT interactions. As described in the literature (J. Lumin. 2013. 136. 296-302 DOI: 10.1016/j.jlumin.2012.12.021 and J. Mol. Struct. 2017. 1128. 636-644. DOI: 10.1016/j.molstruc.2016. 08.081), the fluorescence quantum yield of Couoxo-LD depends on the polarity of the solvent as well as on specific solute-solvent interactions such as hydrogen bonding and high photostability. Thank you again for your positive and constructive suggestions on our manuscript!
- In Figure 7, staining of LDs in The HeLa cells. Why both BODIPY and the Couoxo-LDshowed nucleus membrane staining which is as strong as LDs staining. Authors showed the intensity pattern in 7e, the position of the arrow is not correct it should be through the lipid droplets, but the entire arrow is on the nucleus membrane. Authors needs to comment and revised it properly. My suggestion is using pre-adipocyte cells to the confirm the LDs staining. Or the treatment of oleic acid to increase the LDs in the cytoplasm.
Reply: Thanks for reviewer’s valuable comments. Based on your suggestions, we have re-supplemented the co-localization experiment of BODIPY and Couoxo-LD (Figure 6). Before testing the experiment, we added oleic acid to stimulate the cells to produce more LDs and then co-incubated Hela cells with both dyes. From the results, they exhibited favorable localization effect on LDs (Figure 6f). In previous tests, there appeared to be staining of the incoming nuclear membrane, which may be due to the poor state of the cells under test. The cellular regions through which the intensity curves passed were reworked and marked with arrows, and an intensity scatter plot of the two dyes was added (Figure 6e). Please see the Fig. 6 on line139 and paragraph 3.7 in the revised manuscript. Thank you again for your positive and constructive suggestions on our manuscript!
Fig. 6. Fluorescence images of live HeLa cells incubated with Couoxo-LD (10 µM) and BODIPY (5 µM) for 30 min at 37 degrees celsius. (a) Bright-field image. (b) Emission of BODIPY (λex = 405 nm , 480–510 nm). (c) Emission of Couoxo-LD (λex = 405 nm , 600–650 nm). (d) Merged images. (e)The intensity scatter plot of (b) and (c). (f) Fluorescence intensity distribution within the rectangle.
- What is the concentration used for BODIPY staining, should be mentioned in the Figure 7 caption. Authors used 10 micromolar for Couoxo-LDfor imaging, did authors try other smaller range of concentrations. Since Couoxo-LDshowed very poor fluorescence in nonpolar media which is not great/useful for bright staining of lipid droplets.
Reply: Thanks for reviewer’s valuable comments. Based on your suggestions, we have increased the concentration of BODIPY. Before performing imaging, we have tried imaging with lower concentrations (2μM, 5μM) of Couoxo-LD before performing imaging, but they were not effective for imaging LDs (Fig S7), In the end, we chose 10μM as the concentration for this bioimaging. The specific test results are shown below and are reflected in the manuscript. Please see Fig. S7, page S6 in the revised supporting information. Thank you again for your positive and constructive suggestions on our manuscript!
Fig. S7. Fluorescence images of live HeLa cells incubated with different concentrations of Couoxo-LD at 37 degrees for 30 minutes.
- Abs. and emission spectra of Couoxo-LDand BODIPY shoud be presented in the one Figure in the manuscript or in the supporting information which will be clear to the biosensor readership that BODIPY and Couoxo-LDcan be used together in microscopy for live cell imaging.
Reply: Thanks for reviewer’s valuable comments. Based on your suggestions, we have supplemented the absorption and emission spectra of Couoxo-LD and BODIPY, as shown in Figure S6 below. We have briefly described them and added them to the supporting material. Please see line 270, paragraph 3.7, page 9 in the revised manuscript, and Fig. S6, page S6 in the revised supporting information. Thank you again for your positive and constructive suggestions on our manuscript!
Fig. S6. Absorption and emission spectra of Couoxo-LD (solid line) and BODIPY (dashed line) in dioxane.
- In Living zebra fish LDs imaging BODIPY should be use as a positive control.
Reply: Thanks for reviewer’s valuable comments. Based on your suggestions, we made a supplement to BODIPY imaging in live zebrafish LDs imaging. As shown in Figure S8 , both dyes stain the zebrafish yolk sac. Please see Fig. S8, page S7 in the revised Supporting Information. Thank you again for your positive and constructive suggestions on our manuscript!
Fig. S8. Fluorescence images of live zebrafish treated with Couoxo-LD (10 μM) and BODIPY (5 μM). (a,d) Bright field of view; (b) red channel, λex = 405 nm, λem = 600 nm-650 nm; (e) green channel, λex = 405 nm, λem = 480 nm-510 nm; (c,f) merged images. Scale bar. 500 μm.
7.Remove the word Novel from the conclusion.
Reply: Thanks for reviewer’s valuable comments. Based on your suggestions, we have changed the word "Novel" to "New" in the conclusion of the manuscript and similar cases that had occurred. Thank you again for your positive and constructive suggestions on our manuscript!
Jian-Yong Wang, Ph.D
Associate Professor
School of Light Industry and Engineering,
Qi Lu University of Technology (Shandong Academy of Sciences),
Jinan, Shandong 250353, P. R., China
Email: wjy@qlu.edu.cn

Reviewer 3 Report
The manuscript by Wang et al. describes the using of a new fluorescent dye which is viscosity and polarity sensitive and has been studied in the context of lipid droplet staining, both in vitro and in vivo. While this new dye appears to have great utility in such applications and could be an important advancement in the field. One of key issues in the development of a fluorescent dye for lipid droplets is that the dye should enable multimodal imaging together with existing two commercially available dyes—BODIPY 493/503 and Nile Red. I like to recommend its publication after some decisive and major revisions commented below.
1. Authors should remove the word “Novel” or change it to “New” and remove the word “Strong” from the title. It better to revise the title like “Viscosity-sensitive Solvatochromic Fluorescent Probes for Lipid Droplets Staining’
2. Solvatochromic compounds generally shows the high quantum yields in nonpolar solvent and low quantum yields in polar solvents. But in table 1 the trend is opposite ΦF in Toluene is 5.2% while in DMSO and DMF is ~11%. Authors should explain and describe in the manuscript?
3. In figure 7, staining of LDs in The HeLa cells. Why both BODIPY and the Couoxo-LD showed nucleus membrane staining which is as strong as LDs staining. Authors showed the intensity pattern in 7e, the position of the arrow is not correct it should be through the lipid droplets, but the entire arrow is on the nucleus membrane. Authors needs to comment and revised it properly. My suggestion is using pre-adipocyte cells to the confirm the LDs staining. Or the treatment of oleic acid to increase the LDs in the cytoplasm.
4. What is the concentration used for BODIPY staining, should be mentioned in the figure 7 caption. Authors used 10 micromolar for Couoxo-LD for imaging, did authors try other smaller range of concentrations. Since Couoxo-LD showed very poor fluorescence in nonpolar media which is not great/useful for bright staining of lipid droplets.
5. Abs. and emission spectra of Couoxo-LD and BODIPY shoud be presented in the one figure in the manuscript or in the supporting information which will be clear to the biosensor readership that BODIPY and Couoxo-LD can be used together in microscopy for live cell imaging.
6. In Living zebra fish LDs imaging BODIPY should be use as a positive control.
7. Remove the word Novel from the conclusion.
Author Response

(The authors gave the same response as above.)

Round 2
Reviewer 3 Report
Authors have now ressolved all the comments/issues. I have no further comments and manuscript can be consider for publication in present form.